# Quantifying Genetic Parameters for Blackleg Resistance in Rapeseed: A Comparative Study

**DOI:** 10.3390/plants13192710

**Published:** 2024-09-27

**Authors:** Jan Bocianowski, Ewa Starosta, Tomasz Jamruszka, Justyna Szwarc, Małgorzata Jędryczka, Magdalena Grynia, Janetta Niemann

**Affiliations:** 1Department of Mathematical and Statistical Methods, Poznań University of Life Sciences, Wojska Polskiego 28, 60-627 Poznań, Poland; 2Department of Genetics and Plant Breeding, Poznań University of Life Sciences, Dojazd 11, 60-632 Poznań, Poland; ewa.starosta@up.poznan.pl (E.S.); tomasz.jamruszka@up.poznan.pl (T.J.); justyna.szwarc@up.poznan.pl (J.S.); 3Institute of Plant Genetics of the Polish Academy of Sciences, Strzeszyńska 34, 60-479 Poznań, Poland; mjed@igr.poznan.pl; 4IHAR Group, Borowo Department, Strzelce Plant Breeding Ltd., Borowo 35, 64-020 Czempiń, Poland; m_grynia@hr-strzelce.pl

**Keywords:** additive effect, epistasis, three-way epistasis, blackleg resistance, *Leptosphaeria* spp., molecular markers, SNP, DArTseq, next-generation sequencing

## Abstract

Selection is a fundamental part of the plant breeding process, enabling the identification and development of varieties with desirable traits. Thanks to advances in genetics and biotechnology, the selection process has become more precise and efficient, resulting in faster breeding progress and better adaptation of crops to environmental challenges. Genetic parameters related to gene additivity and epistasis play a key role and can influence decisions on the suitability of breeding material. In this study, 188 rapeseed doubled haploid lines were assessed in field conditions for resistance to *Leptosphaeria* spp. Through next-generation sequencing, a total of 133,764 molecular markers (96,121 SilicoDArT and 37,643 SNP) were obtained. The similarity of the DH lines at the phenotypic and genetic levels was calculated. The results indicate that the similarity at the phenotypic level was markedly different from the similarity at the genetic level. Genetic parameters related to additive gene action effects and epistasis (double and triple) were calculated using two methods: based on phenotypic observations only and using molecular marker observations. All evaluated genetic parameters (additive, additive-additive and additive-additive-additive) were statistically significant for both estimation methods. The parameters associated with the interaction (double and triple) had opposite signs depending on the estimation method.

## 1. Introduction

Plant breeding plays a crucial role in agriculture by providing varieties with improved yield, disease resistance, and adaptation to different environmental conditions [1]. A central aspect of this process is selection, which allows for the identification and development of genotypes with desirable traits [2,3]. Modern selection methods, supported by advanced genetic technologies, have become more precise and efficient, contributing to significant progress in breeding [4,5]. Selection in plant breeding involves choosing genotypes with desired traits and further propagating them, leading to the improvement of the breeding population [6]. This process is based on the assumption that certain phenotypic traits are inherited and can be enhanced through an appropriate crossing of selected individuals [7,8]. Selection is a key mechanism for improving traits such as yield, product quality, resistance to abiotic and biotic stresses, and adaptation to changing climatic conditions [9,10]. The history of selection in plant breeding dates back to the beginning of agriculture, when farmers intuitively selected the best plants for propagation. Over time, this process was refined by introducing systematic selection methods based on Mendelian genetics principles. Modern approaches to selection, such as Marker-Assisted Selection (MAS) and genomic selection, have significantly increased the precision and efficiency of breeding [11,12]. Mass selection involves selecting plants with desirable traits from a population and using their seeds to create the next generation [13]. This method is relatively simple and inexpensive, but its effectiveness may be limited for low heritability traits or complex genetic architecture. Pedigree selection involves tracking the lineage of selected plants over several generations [14,15]. This allows for a better understanding of how traits are inherited and enables more precise selection. This method is particularly useful in the breeding of polygenic traits [16]. Inbreeding line selection involves creating lines of plants with a high degree of homozygosity through repeated crossing within the same line [17,18]. This selection method allows for the stabilization of desirable traits and is crucial in the breeding of pure (homozygous) varieties [19]. MAS allows for faster and more precise selection [20]. Genomic selection is the most advanced form of selection, using full genotype profiles to predict the breeding value of plants [21,22]. This enables the selection of plants with desirable traits already at the seedling stage, significantly accelerating the breeding process. Genomic selection is particularly useful in the breeding of polygenic traits and traits with low heritability [23,24]. One of the main challenges in selection is the complexity of polygenic traits, which are controlled by many genes and can be strongly influenced by the environment. Precise selection of such traits requires advanced genetic and statistical methods [25,26]. Selection carried out over many generations can lead to a reduction in genetic diversity within breeding populations [27]. A narrow genetic pool can increase the risk of losing valuable adaptive traits and resistance to new pathogens [28]. Therefore, it is important to maintain adequate genetic diversity by incorporating wild relatives and diverse genetic materials into breeding programs.

The breeding process can be significantly shortened by using doubled haploid (DH) lines, which are one of the most important tools in modern plant breeding [29,30]. Due to their genetic stability and ability to rapidly achieve homozygosity, DH lines also enhance the precision of selecting desired traits [31,32,33]. Doubled haploid lines are fully homozygous plants created by doubling the chromosome number in haploid gametes (e.g., microspores or megaspores). This process results in lines with a constant, stable genetic structure, which is crucial for many breeding applications. Thanks to this stability, DH lines enable precise selection and predictable inheritance of traits in subsequent generations [34]. In plant breeding, DH lines are used for many purposes, including accelerating the development of new varieties, gene mapping, studying genetic interactions, and producing hybrid varieties [35]. Because these lines are fully homozygous, genetic variation is eliminated, making it easier to identify and select desirable traits. One of the main benefits of their use is the significant reduction in time required to introduce new varieties. The traditional breeding process requires many generations to achieve full homozygosity. With DH lines, fully homozygous material can be obtained in a single generation, speeding up selection and the introduction of new varieties to the market. They are also invaluable in genetic research, especially in gene mapping and the identification of quantitative trait loci (QTL). With full homozygosity, homozygous eliminates the issue of allele segregation, facilitating precise mapping and analysis of trait inheritance. DH lines are widely used in the production of hybrid varieties, where genetic stability is crucial. Fully homozygous DH lines can be used as parents in crosses, ensuring high genetic uniformity of hybrids and predictable trait expression. They are also a valuable tool in studies of genetic interactions, including epistasis and intergenic interactions. The genetic stability of DH lines allows for precise investigation of the effects of individual genes on phenotypic traits [36,37]. The use of DH lines has the potential for further development in combination with modern genetic technologies, such as genome editing (e.g., CRISPR/Cas9) and functional genomics. Integrating these technologies can lead to even faster and more precise creation of plant varieties with desired traits. DH lines can play a key role in precision breeding, where plant genotyping is used to select genotypes with optimal traits for specific environmental conditions. Thanks to their stability, these plants can be an ideal starting material for such advanced breeding programs [38].

The effectiveness of selection depends, among other factors, on the breeder’s decisions, which are based on accurate and reliable information about the genetic traits of the plants [39,40]. In this context, genetic parameters related to the additive effects of genes and epistasis play a crucial role and can influence decisions regarding the suitability of breeding material. Genetic variance is a measure of trait variability in a population that is caused by genetic differences between individuals [41]. It forms the basis of selection, as these genetic differences enable breeders to choose plants with desirable traits. The value of genetic variance determines how quickly and effectively a particular trait can be improved in a population [27,42]. If the research material is homozygous (DH lines, inbred lines), the genetic variance consists of additive and epistatic components [43,44,45]. Additive variance accounts for the effects of genes acting in an additive manner, meaning the effects of these genes sum up. Additive variance is particularly important in selection because it is easily inherited and accounts for changes in the average trait values in subsequent generations. Epistatic variance arises from interactions between different genes [46,47]. Epistasis can significantly influence the phenotype, but its complexity makes it difficult to predict the outcomes of selection [48,49]. Recently, attention has been drawn to triple interactions of QTL-QTL-QTL gene actions and their significant role in shaping the expression of quantitative traits [50,51,52,53,54]. High genetic variance in a population enables effective selection because there is greater phenotypic diversity from which to choose plants with desirable traits. In particular, additive variance is crucial, as it directly impacts the effectiveness of selection [55].

An earlier study [56] showed that the main effects of genotype were important for the degree of resistance to blackleg. Blackleg, caused by *Leptosphaeria maculans* and *Leptosphaeria biglobosa* fungal complex, is one of the most economically important diseases of rapeseed [57,58]. Increasing canola production and seed oil demand exacerbate the expansion and severity of the disease. The symptoms of infection manifest in the form of leaf, stem, and pod lesions. The most severe symptom is the formation of stem canker in plants which results in limited nutrient transport, and therefore yield loss [59]. Each year the incidence of blackleg worldwide is 10%. In some cases, losses reached 50% [60,61]. Blackleg can be managed in several ways such as crop rotation, tillage, and fungicidal treatments [62]. However, the most effective, environmentally safe, and economical method is the implementation of resistant varieties of rapeseed. Up to date, there are over 20 known major blackleg resistance genes (*R* genes) [63,64,65]. However, it was reported that the effectiveness of *R* genes can be quickly overcome by rapidly adapting pathogens [64]. Several Quantitative Trait Loci have been reported, but their validity is hard to attest due to strong genotype × environment interactions [65]. Currently, studies are highly focused on the development of molecular markers linked to blackleg resistance in rapeseed [66]. These markers may aid the breeding process and hasten the development of new resistant cultivars. Sadly, these markers are rarely used in plant selection, due to variable rapeseed germplasm. Breeding protocols usually rely on phenotyping using differential isolates of *Leptosphaeria maculans* [67].

The differences between the mean values of the DH lines were so large that it was decided to further test the structure of their differentiation phenotypically as well as genetically. The objectives of this study were (i) to assess the similarity of 188 DH lines at the phenotypic and genetic levels together with hierarchical clustering, and (ii) to estimate genetic parameters related to additive gene action, additive-additive interaction, and additive-additive-additive interaction using two methods: based solely on phenotypic observations and also using observations of molecular markers. Following this approach, the difference between clustering genotypes using molecular and phenotypic markers was revealed, and the statistical significance of the genetic parameters was determined.

## 2. Results

### 2.1. Phenotyping of Rapeseed DH Lines for Resistance to Blackleg

Analysis of variance indicated that the main effects of genotype were significant for degree of resistance to blackleg [56]. The differences between the DH lines were so great that it was decided to further check the structure of the differentiation.

The similarity of DH lines in terms of blackleg (*Leptosphaeria* spp.) resistance calculated from formula (2) ranged from 0 (for five pairs of lines) to 1 (for 832 pairs of DH lines), with a mean of 0.837. A dendrogram showing phenotypic similarity among the 188 DH lines analyzed was constructed from similarity coefficients (2) using the UPGMA method of blackleg (*Leptosphaeria* spp.) resistance observations (Figure 1). The dendrogram distinguished five similarity groups containing four, 35, 58, 45 and 46 DH lines, respectively (Figure 1).

Estimates of all three genetic parameters calculated from observations of phenotypic blackleg (*Leptosphaeria* spp.) resistance only were statistically significant. The value of the parameter associated with the additive effect of genes was equal to 2.450. The estimated value of epistasis was equal to 0.435. In contrast, the effect of triple gene-gene-gene interaction was 0.952 (Table 1).

### 2.2. Genotyping of Rapeseed DH Lines

A total of 133,764 molecular markers (96,121 SilicoDArT and 37,643 SNP) were obtained. Based on these markers, a dendrogram of genetic similarity between the 188 DH lines was constructed. Six groups of similarity were distinguished on the dendrogram (Figure 2). The grouping of DH lines at the genetic level did not coincide with the grouping at the phenotypic level. The correlation coefficient between the two similarities was equal to 0.0266. Although this was a statistical significance of 0.001 (with the number of degrees of freedom equal to 17,576), no correlation can be said in this case.

Of the 15 molecular markers associated with resistance to blackleg (*Leptosphaeria* spp.) selected by association mapping in an earlier study [56], five were selected (by means of backward step selection) for a multiple model with significant additive main effects (Table 2). In the aforementioned study, it was found that three of those markers were located within gene sequences. It was determined that their expression levels after *L. maculans* inoculation changed significantly. Furthermore, the putative function of these genes was extensively described based on their orthologues in *Arabidopsis thaliana* model species: marker 7853—sequence localized within 13th exon of *BnaA06g11460D*, marker 11720—SNP localized within the 12th (last) exon of *BnaC06g36400D*, marker 12134—SNP localized between *BnaC06g42650D* (1255 bp from START codon) and *BnaC06g42660D* (33,148 bp from STOP codon), marker 12232—SNP localized within the 9th exon of *BnaC03g02160D*, and marker 12456—SNP localized within the 1st intron of *BnaA08g17000D*.

These five markers associated with QTLs determining the observed trait were used to construct a multiple regression model including main effects, double interaction effects of all QTL-QTL pairs, and interaction effects of all QTL-QTL-QTL triplets. The final regression model was of the form:(1)$$y=1.287+0.776\cdot m_{12134}+0.548{\cdot m}_{12232}+\;+0.470\cdot m_{12456}-0.532\cdot m_{11720}\cdot m_{12134}-0.478\cdot m_{7853}\cdot m_{12134}\cdot m_{12232}.$$

The final model (significant at the 0.001 level) explained 19.3% of the percentage of variation in blackleg (*Leptosphaeria* spp.) resistance. Estimates of all three genetic parameters calculated using molecular marker observations were statistically significant, as were those based on phenotypic observations only (Table 1). The total additive effect of genes was equal to 1.794. The interaction-related parameters had negative signs, in contrast to the phenotypic assessments, and were −0.532 and −0.478 for double interaction (epistasis) and triple interaction, respectively (Table 1).

## 3. Discussion

The selection of breeding materials is the foundation of the plant breeding process, allowing for the development of varieties with desirable traits [5]. As agriculture faces increasing challenges such as climate change, growing food demand, and limited natural resources, effective selection methods are becoming crucial for the development of new crop varieties [68,69]. The lack of concordance between phenotypic similarity and genetic similarity complicates the selection process. Kozak et al. [70] previously noted that genetic diversity is not the same as phenotypic diversity. In the presented studies, this observation was confirmed by obtaining a linear correlation coefficient between phenotypic and genetic diversity equal to *r* = 0.0266. Seyis et al. [71] analyzed the genetic diversity of resynthesized rapeseed generated from interspecific hybridization between suitable forms of *Brassica rapa* L. (syn. *campestris*) and *B. oleracea* L. and obtained results similar to those in this study.

The different clustering of DH lines at the phenotypic and genetic levels was likely the cause of the varying assessments of the genetic parameters evaluated using the two proposed methods. Other researchers have reached similar conclusions: Hannan et al. [72], examining 43 rice genotypes; and Chen et al. [73] for 258 DH lines derivatives of a cross between a canola variety Quantum and a resynthesized *B. napus* line No. 2127-17, as well as a fixed immortalized F_2_ population generated by randomly permuted intermating of these DHs. This was particularly significant for parameters related to gene interactions determining blackleg (*Leptosphaeria* spp.) resistance. The calculated assessments had opposite signs (which is not an issue, as this can be adjusted during the validation of selected markers) and showed considerable differences in values. Epistasis plays a crucial role in shaping the phenotypic traits of plants [74,75]. It is a phenomenon where one gene can mask or modify the effect of another gene, leading to complex patterns of inheritance and phenotypic variability [76]. Understanding and properly utilizing epistasis in plant breeding is essential for the effective selection and development of varieties with desirable traits [45]. Epistasis complicates classical inheritance patterns, which are based on Mendelian principles [77]. Since epistatic interactions can determine trait values, predicting phenotypes based on genotypes becomes more challenging [78,79]. This can lead to unexpected outcomes in breeding, where traits may not be inherited as expected. Selection in the presence of epistasis requires advanced strategies that account for gene interactions [80]. In traditional selection methods, which focus on individual genes, the effects of epistasis may be overlooked, leading to incomplete control over trait inheritance. Modern methods, such as genomic selection, allow for more precise management of epistasis in the breeding process [81,82,83].

The effects of additive gene action and epistasis for blackleg resistance in rapeseed were also evaluated by other authors [84,85,86,87,88]. Pang and Halloran [84] studied the genetic control of adult-plant blackleg [*Leptosphaeria maculans* (Desm.) Ces. et De Not.] resistance in rapeseed (*B. napus* L.) in the F_2_ and first-backcross populations. Kumar et al. [85] used three segregating populations derived from the resistant cv. Darmor and multi-year data available on oilseed rape panels to obtain a wide overview of the genomic regions involved in quantitative resistance to blackleg in oilseed rape. Pilet et al. [86] analyzed 152 DH lines derived from the F1‘Darmor-bzh’×‘Yudal’ and obtained ten QTLs with significant additive effects. Zhao and Meng [87] studied a rapeseed population of 128-F2:3 families derived from a cross between the male sterility restorer line H5200 and a partial resistant line Ning RS-1 and obtained three QTLs with additive effect. Additionally, the authors observed epistasis effects for the resistance. Zhao and Meng [87] concluded that both single-locus QTLs and epistatic interactions played important roles in *Sclerotinia* resistance in rapeseed. Larkan et al. [88] obtained significant additive and additive-additive interaction effects for a population of 242 DH lines.

Genetic interactions play a crucial role in plant breeding, shaping the expression of phenotypic traits and determining the success of selecting genotypes with desirable characteristics [89]. While classical epistatic interactions involve interactions between two genes, more complex interactions, such as triple interactions that involve the influence of three different genes on a single phenotypic trait, are gaining increasing attention in breeding research [50,51]. Understanding these complex interactions is key to effective selection and development of plant varieties with optimal characteristics. Triple interaction refers to a situation where three different genes interact to determine a phenotype. Compared to simple two-gene interactions, triple interactions can lead to much more complex inheritance patterns, where the effect of one gene is modified by the presence of two other genes [52,53,54]. This can result in unexpected phenotypes that are not predictable based on the interaction of individual genes alone. In plant breeding, triple interactions can play a crucial role in determining complex quantitative traits. Understanding these interactions will enable breeders to more precisely select genotypes that exhibit synergistic gene effects, leading to a higher level of desirable traits. Triple interactions may also contribute to the stabilization of phenotypic traits across different environmental conditions [54]. When traits are controlled by more than two genes, their expression may be more stable and less susceptible to environmental changes. Breeders can utilize this stability to select plants that exhibit consistent traits regardless of changing climatic conditions. Analyzing triple interactions is significantly more complex than analyzing simple two-gene interactions. It requires advanced statistical and bioinformatics methods that can account for multi-level interactions. Traditional genetic analysis methods may be insufficient to fully understand these phenomena.

## 4. Materials and Methods

### 4.1. Plant Material

188 doubled haploid (DH) lines of rapeseed derived from Strzelce Plant Breeding Ltd. (Strzelce, Poland) IHAR Group, Borowo were used as a plant material in this study. To ensure that DH lines would represent a variety of disease response structures in the mapping population, intraspecific crossings of registered rapeseed varieties with confirmed blackleg resistance were performed.

### 4.2. Field Assessment

All of the doubled haploid lines were subjected to in-field assessment of blackleg resistance, conducted on testing fields belonging to Strzelce Plant Breeding Ltd. IHAR Group, Borowo (52,11850° N, 16,78870° E), Poland. The experiment was established in a randomized complete block design in three replicates. The level of blackleg resistance was evaluated during the BBCH 70–89 plant growth phase (Autumn 2023), by the use of a 0–9 symptoms severity scale by Jędryczka [90], which was described in detail in a previous study by Starosta et al. [56].

### 4.3. DNA Extraction and Genotyping

Seeds of the 188 DH lines were sown on sterile Petri dishes to obtain young seedlings, from which whole genomic DNA was isolated using a Genomic Mini AX Plant kit (A&A Biotechnology, Gdańsk, Poland). The concentration and purity of the DNA samples were tested and equal dilutions to 100 ng µL^−1^ were prepared. After placing in two 96-well plates, the samples were sent to Diversity Arrays Technology (University of Canberra, Australia) for DArTseq analysis. The latter procedure was a part of the preceding study by Starosta et al. [56]. Briefly, it includes steps such as reducing the genome complexity, PCR amplification, and Illumina sequencing followed by filtering and identification of markers. Through next-generation sequencing, a total of 133,764 molecular markers (96,121 SilicoDArT and 37,643 SNP) were obtained.

### 4.4. Phenotypic and Genotypic Similarity

Phenotypic similarity ($s_{P}$) of genotypes in terms of blackleg (*Leptosphaeria* spp.) resistance was calculated based on the proposed formula:(2)$$S_{P,i,j}=1-\frac{\left|{\bar{y}}_{i}-{\bar{y}}_{j}\right|}{max\left({\bar{y}}_{\cdot }\right)},$$
where $S_{P,i,j}$ denotes the phenotypic similarity between line *i* and line *j*, ${\bar{y}}_{i}$ denotes the mean value of blackleg resistance of the *i*-th line, ${\bar{y}}_{j}$ denotes the mean value of blackleg resistance of the *j*-th line, and $max\left({\bar{y}}_{\cdot }\right)$ denotes the highest mean value of blackleg resistance among the DH lines tested.

Genetic similarity among 188 rapeseed DH lines was estimated based on molecular marker observations by Nei’s measure [91,92]:(3)$$S_{G,i,j}=\frac{2N_{ij}}{N_{i}+N_{j}},$$
where $S_{G,i,j}$ denotes the genetic similarity between line *i* and line *j*, $N_{i}$ denotes the number of alleles present in *i*-th DH line, $N_{j}$ denotes the number of alleles present in *j*-th DH line, and $N_{ij}$ denotes the number of alleles present in both line *i* and line *j*.

The calculated similarity coefficients, phenotypic and genotypic, were used to group the DH lines using the unweighted pair group method with arithmetic mean (UPGMA) method. The results of the groupings (based on phenotypes and genotypes, independently) were presented as dendrograms.

### 4.5. Genetic Parameters

#### 4.5.1. Estimation Based on the Phenotype

The total additive $\left(a_{P}\right)$ gene effect on the basis of phenotypic observations of blackleg (*Leptosphaeria* spp.) resistance can be estimated by the following formula [93]:(4)$$\hat{a_{P}}=\frac{1}{2}\left({\bar{L}}_{max}-{\bar{L}}_{min}\right),$$
where ${\bar{L}}_{max}$ and ${\bar{L}}_{min}$ are the means for the extreme groups (maximal and minimal lines, respectively) [94]. Groups of extreme lines were identified by the quantile method [93] in which lines with mean values smaller (bigger) than 0.03 (0.97) quantile of the empirical distribution of means are assumed as minimal (maximal) lines.

The total additive by additive $\left({aa}_{P}\right)$ epistasis interaction effect on the basis of phenotypic observations of blackleg resistance can be estimated by the following formula [95,96]:(5)$$\hat{{aa}_{P}}=\frac{1}{2}\left({\bar{L}}_{max}+{\bar{L}}_{min}\right)-\bar{L},$$
where $\bar{L}$ is the mean of all DH lines.

The total additive by additive by additive $\left({aaa}_{P}\right)$ three-way interaction effect on the basis of phenotypic observations of blackleg (*Leptosphaeria* spp.) resistance can be estimated by the following formula [50]:(6)$$\hat{{aaa}_{P}}=\frac{1}{2}\left({\overset{̿}{L}}_{max}+{\overset{̿}{L}}_{min}\right)-\bar{L},$$
where ${\overset{̿}{L}}_{max}$ and ${\overset{̿}{L}}_{min}$ are the means for 0.99 and 0.01 quantile, respectively, DH lines.

The test statistics to verify hypotheses about genetic parameters different than zero are given by:(7)$$F_{a_{P}}=\frac{{MS}_{a_{P}}}{{MS}_{e}},\;F_{{aa}_{P}}=\frac{{MS}_{{aa}_{P}}}{{MS}_{e}}\;\mathrm{and}\;F_{{aaa}_{P}}=\frac{{MS}_{a{aa}_{P}}}{{MS}_{e}},$$
where ${MS}_{a_{P}}$, ${MS}_{{aa}_{P}}$, ${MS}_{{aaa}_{P}}$, and ${MS}_{e}$ are mean squares for parameters $a_{P}$, ${aa}_{P}$, ${aaa}_{P}$, and residual, respectively.

#### 4.5.2. Estimation Based on the Genotypic Observations

The additive $\left(a_{G}\right)$ QTL effect, additive by additive $\left({aa}_{G}\right)$ epistasis QTL-QTL interaction effect, and additive by additive by additive $\left({aaa}_{G}\right)$ three-way QTL-QTL-QTL interaction effect were estimated based on the methods proposed by Bocianowski and Krajewski [93], Bocianowski [95], and Cyplik and Bocianowski [50], respectively. In an earlier study [56], 15 molecular markers (nine SilicoDArT and six SNPs) associated with blackleg (*Leptosphaeria* spp.) resistance were selected based on association mapping. These markers, associated with QTLs determining the observed trait, were used to construct a multiple regression model including main effects, double interaction effects of all QTL-QTL pairs, and interaction effects of all QTL-QTL-QTL triplets:(8)$$y_{lijkstu}=\mu +\sum_{i}a_{i}\cdot m_{i}+\sum_{j}\sum_{k}{aa}_{jk}\cdot m_{j}\cdot m_{k}+\sum_{s}\sum_{t}\sum_{u}{aaa}_{stu}\cdot m_{s}\cdot m_{t}\cdot m_{u}+e_{lijkstu},$$
where $y_{lijkstu}$ is the mean value of blackleg (*Leptosphaeria* spp.) resistance for *l*-th DH line, $m_{x}$ denotes observation of *x* marker, and $e_{lijkstu}$ denotes the random component. Markers chosen for model (8) were selected using a stepwise regression procedure [50]. In this case, a two-step algorithm was used as follows: (i) selection of markers with main effects was carried out by backward stepwise search; and (ii) double and triple interactions were considered for markers selected in the first step. The final total additive effect of genes was calculated as the sum of the absolute values of the individual effects [93]. However, the total interaction effects of QTL-QTL and QTL-QTL-QTL genes were calculated as the sum of the individual double and triple interaction effects, respectively.

All analyses were conducted using the statistical package Genstat version 23.1 [97].

## 5. Conclusions

Doubled haploid lines are a key breeding material that significantly accelerates and enhances the selection process in plant breeding. Due to their complete homozygosity and genetic stability, DH lines are invaluable in the creation of new varieties, gene mapping, hybrid production, and genetic research. Despite certain technical challenges and limitations, DH lines have enormous potential in the further development of plant breeding, especially when combined with modern genetic technologies. Contemporary breeding increasingly employs a combination of different methods to optimize the selection process and achieve varieties with desirable traits in a shorter time. Genetic parameters play a crucial role in the selection process of breeding materials. Understanding and properly applying these parameters allows breeders to make more effective and informed selections, which in turn leads to the development of plant varieties with desirable traits. The results obtained indicate the necessity of considering various methods of estimating genetic parameters when making decisions related to the selection of materials in the breeding process. Double interaction (epistasis) and triple interaction of gene actions play a significant role in plant breeding, influencing the inheritance of traits, genetic variability, and the effectiveness of selection. Understanding and utilizing these interactions in breeding processes enables the development of plant varieties with desirable characteristics.

## Figures and Tables

**Figure 1 plants-13-02710-f001:**
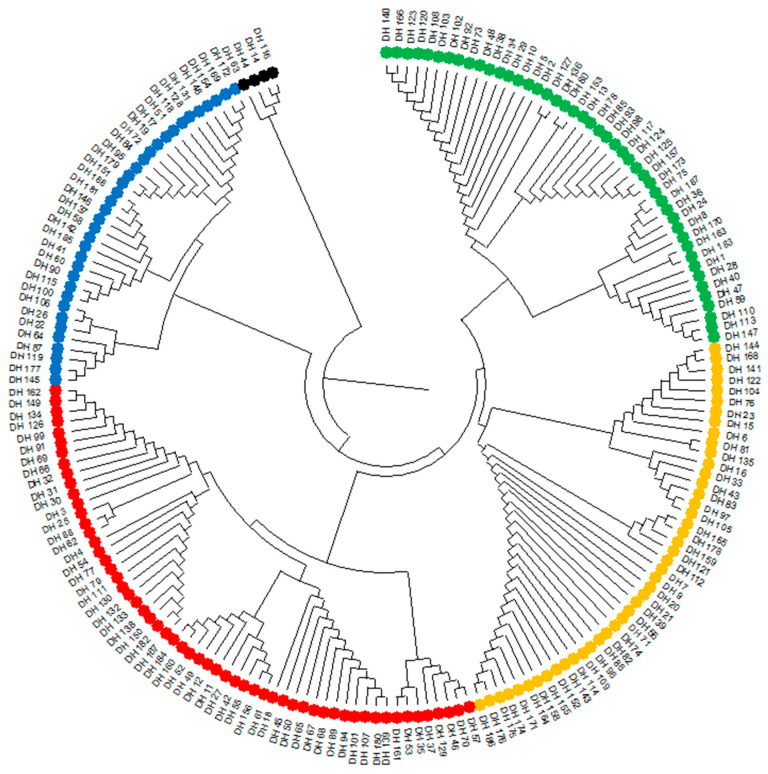
Dendrogram showing phenotypic similarity of blackleg (*Leptosphaeria* spp.) resistance of 188 doubled haploid (DH) lines, constructed from the UPGMA method based on measure (2).

**Figure 2 plants-13-02710-f002:**
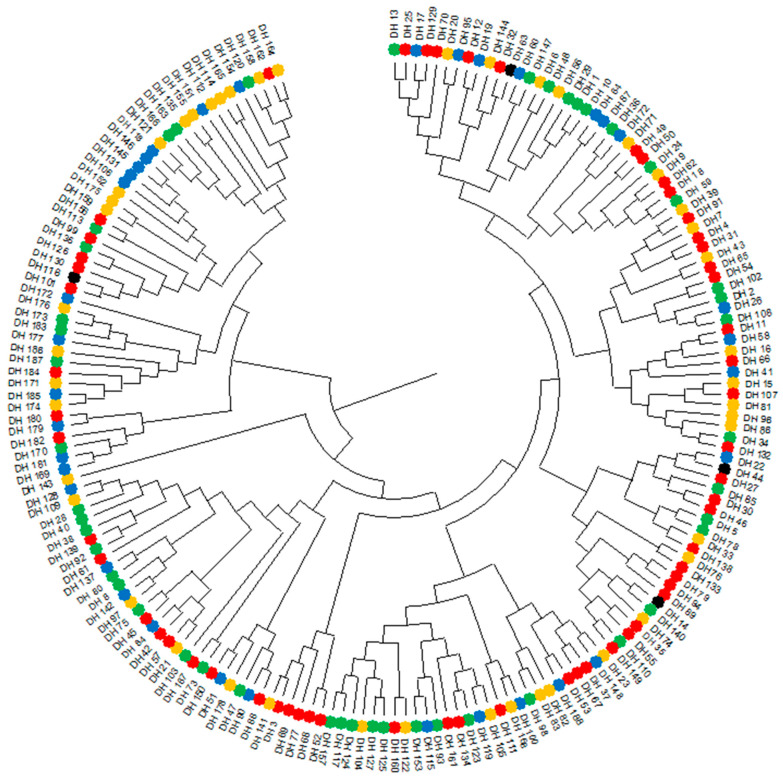
Dendrogram showing the genetic similarity between 188 DH lines constructed based on the observation of 133764 molecular markers. DH lines marked with dots of each color indicate groups formed on a dendrogram constructed from phenotypic observations of blackleg (*Leptosphaeria* spp.) resistance (Figure 1).

**Table 1 plants-13-02710-t001:** The additive, epistasis (additive by additive interaction), and triple interaction (additive by additive by additive) effects for blackleg (*Leptosphaeria* spp.) resistance of 188 DH lines of rapeseed estimated on the basis of phenotypic and genotypic methods.

Genetic Parameter	Phenotypic Method	Genotypic Method
additive, *a*	2.450 ***	1.794 ***
additive-additive, *aa*	0.435 *	−0.532 **
additive-additive-additive, *aaa*	0.952 **	−0.478 *

* *p* < 0.05; ** *p* < 0.01; *** *p* < 0.001.

**Table 2 plants-13-02710-t002:** SilicoDArT and SNP molecular markers significantly associated with resistance of *B. napus* to blackleg [56] selected for a multiple model.

Marker Number	Marker Type	Chromosome	Marker Position on Chromosome (bp)	Marker Sequence
7853	SilicoDArT	A06	5,968,268–5,968,337	TGCAGAAAATGGAATGTTCTTGAGAGATCCTAGTGGAGAATGGGTGACAAATATGCCTCAAGACATGAA
11720	SNP (18:A>T)	C06	34,959,583–34,959,652	TGCAGAAGCAGCCATGAGACAGTATTGCTGTTGAGATATATTGTTGCTGTACCTTGGGGAGGAAGCAAC *
12134	SNP (29:G>A)	C06 random	2,170,846–2,170,777	TGCAGCTTCTACTTTTAGTTGGACAGAGCGCTCAAAGTCAACAATTACAGATCGGAAGAGCGGTTCAGC
12232	SNP (46:T>A)	C03	1,024,177–1,024,246	TGCAGAAAAAGATTCAGGTTCCCGGGACCTGAAGATCACTGGATTGTCTGATGCTGTGTTAGGATGCAT
12456	SNP (45:A>C)	A08	13,650,920–13,650,989	TGCAGTTTCTACACGTACATATCCAATATTTTAGTTTACTTAGGAAGAAATTTGAAATTTGATTTTATT

* SNP positions within the markers are underlined.

## Data Availability

The data presented in this study are available on request from the corresponding author.

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
