# Peer review of "Quantifying Genetic Parameters for Blackleg Resistance in Rapeseed: A Comparative Study"

_plants, 2024, doi:10.3390/plants13192710_

Round 1

Reviewer 1 Report

Comments and Suggestions for Authors

The authors represent a study on the divergent blackleg resistance of rapeseed among about two hundred DH lines. According to the description, the DH lines were derived from Strzelce plant breeding Ltd. and limitedly documented. Comparing to the computational parts, experimental parts are poorly described; for example, I could not specify the cultivation location, cultivation time and season. The main text well reads while other two major questions are raised for further improvement before publication.

(1) Introduction fully covers the concept of related fields but more concise one would be better. The present structure of introduction focuses on the theoretical side. However, the main content of this study is genetic analysis of the DH lines in terms of blackleg resistance. However, nothing was introduced about blackleg resistance. Actually, one paragraph should be made to explain what type of disease blackleg is and what is known about blackleg resistance in genetics. Otherwise readers would not be able to specify the significance of this study. In an alternative word, clarification of the novelty and significant is needed to understand Discussion part completely.

(2) Blackleg resistance-associated DNA markers were listed on Table 2. How are the potentials of functional SNPs?

Comments on the Quality of English Language

None.

Author Response

Response to Reviewer 1 Comments

Reviewer #1

Point 1: The authors represent a study on the divergent blackleg resistance of rapeseed among about two hundred DH lines. According to the description, the DH lines were derived from Strzelce plant breeding Ltd. and limitedly documented. Comparing to the computational parts, experimental parts are poorly described; for example, I could not specify the cultivation location, cultivation time and season. The main text well reads while other two major questions are raised for further improvement before publication.

Response: The authors are grateful for valuable comments of the Reviewer #1. We have corrected the manuscript plants 3207575 according to all suggestions. First of all we would like to explain that the cultivation area was indicated in Chapter 4 titled Materials and Methods, 4.2. Field Assessment. It was mentioned that …”conducted on testing fields belonging to Strzelce Plant Breeding Ltd. IHAR Group Borowo. The experiment was established in randomized complete block design, in three replicates. The level of blackleg resistance was evaluated during the BBCH 70-89 plant growth phase…”. In addition, we added geographic coordinates (52,11850° N, 16,78870° E) and a date for phenotypic observations (Autumn 2023).

Point 2: (1) Introduction fully covers the concept of related fields but more concise one would be better. The present structure of introduction focuses on the theoretical side. However, the main content of this study is genetic analysis of the DH lines in terms of blackleg resistance. However, nothing was introduced about blackleg resistance. Actually, one paragraph should be made to explain what type of disease blackleg is and what is known about blackleg resistance in genetics. Otherwise readers would not be able to specify the significance of this study. In an alternative word, clarification of the novelty and significant is needed to understand Discussion part completely.

Response: We added a short paragraph explaining the importance of the blackleg disease in rapeseed production and the significance of molecular markers in rapeseed breeding: “Blackleg, caused by Leptosphaeria maculans and Leptosphaeria biglobosa fungal complex is one of the most economically important diseases of rapeseed [57,58]. Increasing canola production and seed oil demand exacerbates the expansion and severity of the disease. The symptoms of infection manifest in the form of leaf, stem, and pod lesions. The most severe symptom is the formation of stem canker in plants which results in limited nutrient transport, and therefore yield loss [59]. Each year the incidence if blackleg worldwide is 10%. In some cases, losses reached 50% [60,61]. Blackleg can be managed in several ways such as crop rotation, tillage, and fungicidal treatments [62]. However, most effective, environmentally safe, and economic method is the implementation of resistant varieties of rapeseed. Up to date, there are over 20 known major blackleg resistance genes (R genes) [63-65]. However, it was reported that the effectiveness of R genes can be quickly overcome by rapidly adapting pathogen [64]. Several Quantitative Trait Loci have been reported, but their validity is hard to attest due to strong genotype x environment interactions [65]. Currently, studies are highly focused on the development of molecular markers linked to blackleg resistance in rapeseed [66]. These markers may aid the breeding process and hasten the development of new resistant cultivars. Sadly, these markers are rarely used in plant selection, due to variable rapeseed germplasm. Breeding protocols usually rely on phenotyping using differential isolates of Leptosphaeria maculans [67].”.

Point 3: (2) Blackleg resistance-associated DNA markers were listed on Table 2. How are the potentials of functional SNPs?

Response: The linkage of markers to candidate genes and the annotation of Arabidopsis thaliana orthologous genes was already presented in our previous study (https://doi.org/10.3390/ijms25158415). We would like to avoid repeating the presentation of previously obtained results in this article, hence we decided to add information in the paragraph 2.2 (Genotyping) referring to the previous publication. We added text: “marker 7853 – sequence localized within 13th exon of BnaA06g11460D, marker 11720 – SNP localized within the 12th (last) exon of BnaC06g36400D, marker 12134 – SNP localized between BnaC06g42650D (1255 bp from START codon) and BnaC06g42660D (33148 bp from STOP codon), marker 12232 – SNP localized within the 9th exon of BnaC03g02160D, and marker 12456 – SNP localized within the 1st intron of BnaA08g17000D.”

Reviewer 2 Report

Comments and Suggestions for Authors

The paper presented the analysis of 188 rapeseed DH lines on genetic and phenotypic levels. The authors concluded that there is significant difference of lines between phenotypes and genotypes. 

To improve the paper's readability and clarity, please see the comments below: 

1) Introduction: there are some repetitive sentences and information in the Introduction. Please rewrite it to make it concise and informative. For example, Line 62-64, says the same thing as Line 51-54. Why would you select the blackleg resistance trait in the rapeseed to study? What has been done and what is missing? In addition, in the last paragraph (line 133-137), it should state the summary of the research instead of the objective.

2) Introduction: There is some literatures mentioned in the Discussion that would fit more appropriately in the Introduction. For example, from Line 211.

3) Results: to make it easier for the readers to follow, please consider adding more details of the two sections (2.1 and 2.2) of the Results, i.e Phenotypes of ......

4) Results: Line 144 forward: please add a brief description of the experiments that was conducted (plants used, procedure, etc) to improve readability. Same for the 2.2.

5) Discussion: some of literature review contents can be moved up to the Introduction, and please focus on discussing the result of this manuscript and the potential impact.

Comments on the Quality of English Language

The English is ok, but not concise. 

Author Response

Response to Reviewer 2 Comments

Reviewer #2

Point 1: The paper presented the analysis of 188 rapeseed DH lines on genetic and phenotypic levels. The authors concluded that there is significant difference of lines between phenotypes and genotypes.

Response: Thank you very much.

To improve the paper's readability and clarity, please see the comments below:

Point 2: 1) Introduction: there are some repetitive sentences and information in the Introduction. Please rewrite it to make it concise and informative. For example, Line 62-64, says the same thing as Line 51-54. Why would you select the blackleg resistance trait in the rapeseed to study? What has been done and what is missing? In addition, in the last paragraph (line 133-137), it should state the summary of the research instead of the objective.

Response: The authors are grateful for valuable comments of the Reviewer #2. We have corrected the manuscript plants 3207575 according to all suggestions.

The Introduction section aimed to present and compare some of the most used breeding methods to highlight the importance of novel breeding techniques. That is why Line 51-54 refers to traditional Mass Selection in contrast to modern molecular approach – Marker Assisted Selection – which is described in Line 62-64. We have removed the passage that was inadvertently repeated.

Moreover, in order to improve the Introduction chapter we added a paragraph describing the blackleg disease importance and management in Brassica napus.

In the last paragraph of the Introduction chapter we aimed to show the linkage between this study and our previous extensive study which focused on the identification of novel molecular markers. As the summary of the research is presented earlier in the Abstract as well in the Discussion section, we decided to present the main goals of our research to bring the Reader’s attention to the purpose of the study. However, as suggested by the Reviewer, we expanded the last paragraph of the Introduction, highlighting the results of the research.

Point 3: 2) Introduction: There is some literatures mentioned in the Discussion that would fit more appropriately in the Introduction. For example, from Line 211.

Response: It seems to us that the citations mentioned by the Reviewer mentioned from line 211 fit more into the Discussion than the Introduction. They are a remark that other researchers, like us in the research presented, have drawn specific conclusions. These are the effects of the results obtained and therefore, in our opinion, fit more into the Discussion than the Introduction.

Point 4: 3) Results: to make it easier for the readers to follow, please consider adding more details of the two sections (2.1 and 2.2) of the Results, i.e Phenotypes of ......

Response: We have expanded the headers of sections 2.1 and 2.2 to clarify the section content and improve the readability of the manuscript.

Point 5: 4) Results: Line 144 forward: please add a brief description of the experiments that was conducted (plants used, procedure, etc) to improve readability. Same for the 2.2.

Response: The methods used to conduct the study are presented in the Methods section of the manuscript, which is located after the Discussion section according to the Journal’s preferences.

Point 6: 5) Discussion: some of literature review contents can be moved up to the Introduction, and please focus on discussing the result of this manuscript and the potential impact.

Response: We have supplemented the Introduction with a paragraph that is a literature review on blackleg resistance. We think this may be sufficient as an introduction to the research problem under consideration. At the same time, we perceive the Discussion as a comprehensive and synthetic discussion and comparison of our results with the effects of the work of other researchers. It seems to us that "moving" any passage to the Introduction would make the Discussion incoherent. Therefore, we would opt to leave it in its current version.

Point 7: The English is ok, but not concise.

Response: The English has been revised to be more concise.

Round 2

Reviewer 2 Report

Comments and Suggestions for Authors

The authors has made changes according to the reviewer's suggestion.